# Geographic Variation in Signal Preferences in the Tropical Katydid *Neoconocephalus triops*

**DOI:** 10.3390/biology13121026

**Published:** 2024-12-07

**Authors:** Oliver M. Beckers, Johannes Schul

**Affiliations:** 1Department of Biological Sciences, Murray State University, 1112B Biology Building, Murray, KY 42071, USA; 2Department of Biological Sciences, University of Missouri, 207 Tucker Hall, Columbia, MO 65211, USA; schulj@missouri.edu

**Keywords:** phenotypic variation, communication system, mating calls, species isolation

## Abstract

Many animals communicate to bring the sexes together for reproduction. However, only if male signals and female preferences match, communication will lead to reproduction. Despite this required match between preferences and signals, animal communication systems evolve rapidly. The processes for how a system with a tight match between signal and preference acquires the variation necessary for evolution to proceed are poorly understood and are the focus of this study. We tested signal preferences for two call traits in a katydid species from two tropical populations. The populations differed drastically in their selectivity for pulse rate and call structure; while Puerto Rican females were highly selective for pulse rate, they were not selective for call structure. Costa Rican females displayed the opposite pattern in selectivity for the two call traits. The differences in the preferences could be due to other katydid species calling in the two regions, potentially interfering with the communication system of our study species. Importantly, the reduced selectivity in females allows for the accumulation of variation in call traits necessary for the system to evolve. Thus, our study provides insights in how communication systems could allow for signal variation to happen while the signals match the preferences.

## 1. Introduction

In the context of reproductive communication, signals function in bringing the sexes of the same species together for mating while keeping individuals of different species apart [1,2,3]. Consequently, sender signals and receiver preferences have to match for the communication system to function, seemingly constraining the diversification of these systems [4]. This tight interaction between the two parts of the system has raised fundamental questions, such as which processes generate the variation necessary for communication to diversify and how these systems can evolve while maintaining the match between signal and preference [4].

Geographic variation in communication systems can be an important factor during the evolution of communication systems, e.g., [5,6,7]. Such variation in communication can be caused by selective evolutionary forces. Differences among habitats in, for example, the transmission of the signals (‘sensory drive’, [8]), the presence of eavesdropping predators and parasites [9,10,11], or the masking of the signals by other signaling species, e.g., [12,13,14] can select for geographic variation in signals and preferences. Pleiotropy between non-communication traits and traits related to communication [6], differences in hidden genetic variability or phenotypic plasticity [15], as well as founder and bottleneck effects are potential sources of genetic variation in communication among populations. Of course, these mechanisms are not mutually exclusive.

Geographic variation in communication has been reported in different animals and modalities, such as the local song dialects in diverse groups of birds, e.g., [16,17,18], the mating calls in cricket frogs (*Acris crepitans* [19,20]), the color patterns and preferences in guppies (*Poecilila reticulata* [21,22]), the calls of the variable field cricket (*Gryllus lineaticeps* [11]), and the túngara frog (*Physalaemus pustolosus* [23]), to mention a few. These examples illustrate that variation within species is present and selection of preferences and signals might not necessarily be as strongly stabilizing as previously assumed [4]. Once population differences in signals and preferences have been established, this variation can ultimately lead to speciation [24,25]. It has been proposed that the early stages of speciation are best understood by examining the causes and consequences in variation among geographic distinct populations, e.g., [26,27]. Our study investigates geographic variation in female preferences in the katydid *Neoconocephalus triops*.

The katydid *N. triops* has a wide distribution, ranging from Peru in South America to Ohio, U.S.A [28]. Along this gradient, populations experience different environmental conditions, including constant tropical conditions to highly seasonal temperate conditions. In temperate environments, *N. triops* is bivoltine and each generation has a different call as the result of differing daylengths [29,30]. In the tropics, where the species originated [31,32], animals mate all year long and experience a range of temperatures across the year but produce only one call type. We previously reported that *N. triops* females use the conspecific double-pulse rate for call recognition [30]. The pulse rate preferences of Florida females were closed, i.e., females preferred the conspecific pulse rates and reject pulse rates faster and slower than the temperature-specific pulse rate.

In poikilothermic anurans and insects, preferences and signals change in parallel with ambient temperature (‘temperature coupling’ [33,34]). This change is reversible and necessary for successful communication within the natural range of ambient temperatures [33]. Our study aimed to establish whether two tropical populations (Puerto Rico and Costa Rica) differed in their preferences within a natural range of ambient temperatures. We predicted that the preference functions for pulse rate should be fundamentally consistent across populations and tropical females should exhibit closed preference functions, as reported for Florida animals [30] and that those preferences would vary with ambient temperature. That is, preference functions should be centered at lower pulse rates when tested at low temperatures compared to preferences tested at higher temperatures. Furthermore, the calls of tropical *N. triops* consist of trains of pulses that are rhythmically interrupted, i.e., they are structured in verses [30,31]. Therefore, we predicted that both tropical populations of *N. triops* would display preferences for these versed calls over continuous calls, which would deviate from the reported preferences for Florida females that display no preferences for call structure [30].

## 2. Material and Methods

### 2.1. Animals

We collected between 15 and 25 adult male and 20–30 adult female *N. triops* from populations in the vicinity of Liberia in Costa Rica and Naguabo in Puerto Rico in 2006 and 2007 and brought the animals to the laboratory at the University of Missouri for breeding to start our laboratory populations. We kept animals in an incubator at a relative humidity of 50–70%, a light/dark cycle of 15/9 h and high/low temperatures of 30/20 °C, respectively. We provided apples and puppy chow (Purina, St. Louis, MO, USA) as food, and grass for oviposition [30]. We replaced the grass every 1–2 weeks for a period of 8–10 weeks and checked the grass for deposited eggs. We raised juveniles from the collected eggs using the protocol described in [30]. We reared juveniles to adulthood under the same conditions as described above for the collected animals and tested preferences of F_1_ females from Puerto Rico and Costa Rica. We kept adult male and female katydids in separate incubators to prevent acoustic stimulation of the females by calling males before the experiments.

### 2.2. Phonotaxis Tests

We tested selectivity for double-pulse rate in female *N. triops* from Costa Rica and Puerto Rico at 20 °C and 25 °C ambient temperatures. We used these temperatures in a previous study to compare the temperature dependence of female response to acoustic stimuli [31]. Note that both populations are multivoltine and breed all year in the tropics [32] experiencing this range of temperatures at sunset across the year (www.climate-data.org accessed on 21 August 2024). We started testing females two weeks after they turned adult for up to four weeks. We did not detect any change in their selectivity during the four weeks of testing.

The experimental set-up is described in detail in [30,31]. In brief, we tested female phonotaxis using a walking compensator (Kramer–Kugel [35]) in a temperature-regulated anechoic chamber. At the beginning of a trial, we placed the test female on top of a sphere where she was free to walk in any direction but kept in place by compensatory sphere rotations. We presented acoustic stimuli from one of two loudspeakers (KSN1218C, Motorola, Chicago, IL, USA) located in the animal’s horizontal plane at a distance of 150 cm and separated by 105°. We switched the broadcast to the second speaker after about 60–90 s of phonotaxis so that we could check that the female reliably tracked the stimulus and was not walking into the direction of the first loudspeaker by chance. Stimulus presentation from the second loudspeaker lasted 60–90 s as well. The direction and speed of walking by a female was recorded from the control circuitry. We conducted all experiments in the dark except for an infrared light placed above the animal that was used to monitor the insects’ position [35,36].

### 2.3. Experimental Protocol

We presented each test and control stimulus for 60–90 s from each of the two loudspeakers. At the beginning of a stimulus series, we tested female response to the control stimulus, followed by two to three test stimuli, another control stimulus, two to three test stimuli, and so on, until all stimuli of a series were tested with the female. We allowed a period of 60 s silence after each stimulus was broadcasted from both loudspeaker positions. The tested stimulus series lasted between 30 and 60 min and consisted of up to six test stimuli interspersed by up to four control stimuli. We randomly varied the sequence of stimuli within a series among individual females to prevent order effects on female responses. For more details on the experimental protocol see [36,37]. Each female was only tested either at 20 °C or 25 °C. For the Puerto Rican population, we tested a total of eight females at 20 °C and nine at 25 °C, and for the Costa Rican population a total of ten at 20 °C and twelve at 25 °C.

### 2.4. Stimulation

We used a custom-developed DA-converter/amplifier system with 250 kHz sampling rate and 16-bit resolution to generate the acoustic signals. We used sinusoids of 11 kHz as carrier signal, which corresponds to the dominant frequency of *N. triops* calls [30], to which we applied amplitude modulations representing the temporal pattern, i.e., the pulses, of the stimuli. Calls of *N. triops* consist of double-pulses [29] and females respond equally strong to stimuli with the double-pulses merged to long pulses [30]. Thus, female *N. triops* uses the conspecific rate of the double-pulses rather than the actual double-pulse structure to recognize mating calls [30]. Therefore, we used stimuli that consisted of pulses equivalent to ‘merged double-pulses’ [31] and refer from here on to these merged double-pulses as ‘pulses’ and ‘pulse rate’. In previous experiments, these stimuli were as attractive as natural calls or models with the double-pulse structure [30,31]. We adjusted the pulse and interval duration of our stimuli to generate stimuli that varied in the pulse rate for the experiments. The duration of the pulses and the pulse intervals was adjusted to keep the pulse duty cycle constant at 72%, which corresponded to the duty cycle of natural calls [OMB unpublished data]. Across populations, we used stimuli that ranged between 55.3 and 119.1 pulses/s at 20 °C and between 71.4 and 240.4 pulses/s at 25 °C. At the lowest pulse rate (55.3 pulses/s), the pulse duration was 13 ms and the pulse interval was 5.1 ms and at the highest pulse rate (240.4 pulses/s), the pulse duration was 3.0 ms and the interval duration was 1.16 ms. The calls of *N. triops* from Costa Rica and Puerto Rico are rhythmically interrupted, i.e., arranged in groups of pulses or verses. For our stimuli, we grouped the pulses of Costa Rican stimuli into 650 ms verses that were interrupted by silent pauses of 50 ms duration and those of Puerto Rican stimuli into 950 ms verses with silent pauses of 50 ms, which correspond to the call parameters of the natural calls from these populations [31].

The control stimulus had the temporal pattern and pulse rate of the conspecific call from each population [31], i.e., 110 pulses/s for the Puerto Rican females and 119 pulses/s for the Costa Rican females at 25 °C. The control stimulus for the Puerto Rican females at 20 °C had a rate of 75 pulses/s and that of the Costa Rican females 85 pulses/s. Note that pulse rates of Costa Rican males are about 10% higher than those of Puerto Rican males, explaining the differences in the control pulse rates [31]. Our control stimuli were highly attractive to *N. triops* females, producing responses comparable to those to conspecific calls [30]. We tested the attraction of females from each population to the two call structures by broadcasting the control stimulus with the pulses arranged in verses and as a continuous string of pulses in separate trials. We used the pulse rate of the control stimulus for each population (see above) and tested female preferences for call structure at 25 °C. We adjusted the signal amplitude for all tested stimuli to 80 ± 1 dB peak SPL (re. 20 µPa) at the position of the female 1 cm above the center of the sphere [30] using a sound level meter (BandK 2231, Brüel & Kjaer, Naerum, Denmark) and a ¼″ condenser microphone (40BF, G.R.A.S, Holte, Denmark).

### 2.5. Data Analysis

We quantified female response to stimuli that varied in merged double-pulse rate by calculating a phonotaxis score [36]. The phonotaxis score included three measures indicating the relative strength of phonotaxis. First, the walking speed of the female to a given test stimulus relative to her walking speed during the control stimulus, which described her locomotion activity. Second, the vector length which described the accuracy of the female’s orientation. Third, the angular orientation in response to the test stimulus relative to that during the control stimulus. Phonotaxis scores can range from approximately +1 to −1, representing perfect positive or negative phonotaxis, respectively. Scores close to 0 indicate random or no phonotaxis (for details, see [36]).

### 2.6. Statistics

We compared the phonotaxis scores in response to continuous and versed stimuli for Puerto Rican females and Costa Rican females using paired *t*-tests for each population (JMP version 18 for Mac). We fitted linear, quadratic, and sigmoidal functions to the phontaxis score distribution across pulse rates at each temperature to determine which model best predicted the distribution of the scores. A linear fit would indicate increasing female attraction in either direction and thus directional selection for faster (or slower) pulse rates. A quadratic model would indicate highest phonotaxis scores at intermediate pulse rates and lower scores toward higher and lower rates, resulting in stabilizing selection. A sigmoidal best fit function would indicate that pulse rates faster (or slower) than the intermediate rate would continually elicit strong female attraction. We used the functions ‘AICtab’ from the ‘bbmle’ package (Version 1.0.25) in R (version 4.2.3) to determine the fit of the models. We used Akaike information criterion scores (AIC) and residual standard errors (RSE) to determine the model that best predicted the distribution of the phonotaxis scores at 20 °C and 25 °C for Puerto Rican and Costa Rican females. We used the lowest AIC and RSE values to determine the model that best fit the phonotaxis score distribution. The best-fit model is indicated for each preference function in the figures.

## 3. Results

### 3.1. Preferences for Call Structure

Puerto Rican females were highly attracted to both the continuous call stimulus as well as the stimulus that was structured into verses (paired *t*-test, DF = 8, T-Ratio = 0.85, *p* = 0.42; Figure 1A). In contrast, females from Costa Rica were significantly less attracted to the continuous stimulus compared to the one structured in verses (paired *t*-test, DF = 9, T-stat = −6.52, *p* < 0.0001; Figure 1B).

### 3.2. Preferences for Pulse Rate

At 20 °C ambient temperature, females from Puerto Rico displayed the highest phonotaxis scores at pulse rates of 80 and 91 pulses/second. The scores dropped sharply at higher and lower pulse rates (Figure 2A). A quadratic function approximated the distribution of the phonotaxis scores over the pulse rates the best, suggesting a closed preference function (Table 1).

At 25 °C ambient temperature, Puerto Rican females showed the highest phonotaxis scores at 102 and 120 pulses/s. The scores dropped toward lower and higher pulses rates (Figure 2A gray curve). Again, a quadratic function best approximated the distribution of the data (Table 2).

At 20 °C, females from Costa Rica displayed highest phonotaxis scores at rates of 90 and 95 pulses/s and scores dropped sharply at higher and lower pulse rates (Figure 2B) similar to the preferences of Puerto Rican females. A quadratic function best approximated the distribution of the phonotaxis scores (Table 3).

At 25 °C ambient temperature, phonotaxis scores dropped below pulse rates of 111 pulses/s. However, females showed high phonotaxis scores at 111 pulses/s and the scores remained high at all the other tested higher pulse rates up to 240 pulses/s (Figure 2B). Curve fitting suggested that a sigmoidal function best represented the data distribution (Table 4).

## 4. Discussion

We detected qualitative geographic variation in female preferences between Puerto Rican and Costa Rican females. First, Costa Rican females discriminated against continuous calls, whereas Puerto Rican females did not. Second, preference functions for pulse rates differed qualitatively between populations at 25 °C, with Puerto Rican females displaying closed preference functions and Costa Rican females open ones.

### 4.1. Geographic Variation in Preferences

Pulse rate preferences changed reversibly between 20 °C and 25 °C ambient temperature in both tested populations, whereas the nature of the change was qualitatively different between populations. At 20 °C, both populations had typical band-pass rate preferences, with high responses at intermediate values rolling off to both sides. At 25 °C, the Puerto Rican population maintained this shape of the preference curve, albeit shifted to higher (=faster) pulse rates. In the Costa Rican population, however, the shape of the preference curve changed to that of a high-pass filter, i.e., the responses remained high above a lower cut-off rate (=open preference).

Even pulse rates twice as fast as the natural pulse rate (i.e., 240 pulses/s) still elicited strong responses in Costa Rican females (Figure 2B). As the auditory system of these katydids is not able to resolve rates of 200 Hz and above [38], it is unlikely that behavioral responses would drop at rates above 240 Hz. This open preference at 25 °C of the Costa Rican population is in stark contrast to preferences of the Puerto Rican and Florida population of *N. triops* where rates higher than 120 pulses/s elicited very weak responses (Puerto Rico: Figure 2A; Florida: [30]).

It is unlikely that the switch between open and closed preferences is caused by a lack of temporal resolution in the peripheral and ascending sensory system. Auditory receptor cells and the ascending pathway are well able to resolve and transmit pulse rates up to 160 Hz to the call recognition networks in the brain [38]. Thus, the rejection of rates this high should be possible, as Puerto Rican (this study) and Florida females [30] were clearly able to do so, as indicated by their weak responses to these fast pulse rates.

The neural basis of this switch to an open preference is the loss of the low pass properties of the rate filter, i.e., signals with higher pulse rates now pass through the filter. The band pass properties of *N. triops*’ and other Tettigoniidae’s pulse rate recognition are likely generated through resonance properties of neurons or networks [39,40]. Neurons and networks inherently possess an intrinsic resonance; transient stimulation is followed by periods of reduced and then increased excitability. Successive stimulation during the increased excitability period is more likely to produce a response, thus resulting in a rate preference [41,42]. The fast double-pulse rates of *N. triops* are approaching the limits for resonance-based rate recognition mechanism as the periods of reduced and increased excitability have to fit within one period of 10 ms or less. Given that the double-pulse rate of Costa Rican *N. triops* is higher than that of the Puerto Rican or Florida populations (120 Hz vs. 105–110 Hz at 25 °C), tuning the rate filter to this fast rate may exceed at 25 °C the capacity of the resonance. One possible outcome could be that the inhibitory or hyperpolarizing phase is not effectively stimulated by the ‘too-fast’ pulse rate at 25 °C so that the filter output and phonotaxis remain high. Thus, the switch to open preference at 25 °C of Costa Rican females may be a side effect of their high pulse rate. Other components of the call recognition mechanism of *N. triops* (e.g., selectivity for verse pattern) may prevent fitness costs of this effect (see below). While we interpret the open preference as non-adaptive, it provides a large range of variation necessary for new call values to evolve and persist ultimately providing the communication system an opportunity to evolve without disrupting the match between calls and preferences.

### 4.2. Acoustic Ecology of Preference Evolution

One potential driver for geographic variation could be regional differences in the presence or composition of competitors that impose different selective pressures on traits of the focal species [7]. For example, geographic variation in resource and mate competition between species has led to regional differences in body size in the spadefoot toads *Spea bombifrons* and *S. multiplicata* [43], differences in intraspecific mate competition has led to the evolution of threshold values for morph expression between populations in the dung beetle *Onthophagus taurus* [44], or variation in competition for seeds has led to the divergence of beak sizes in different populations of the ground finches *Geospiza fuliginosa* and *G. fortis* [45]. One important ecological niche is the acoustic space that is used by signalers to communicate with conspecifics and isolate themselves from heterospecifics, e.g., [46]. Competition for acoustic space can lead to differences in signals and preferences, and this kind of competition can lead to geographic variation if the species composition varies among populations [4].

In Costa Rica, *N. triops* occurs sympatrically with *N. punctipes* and *N. maxillosus* [29,32]. *Neoconocephalus punctipes* produce continuous, fast calls with pulse rates of 240–250 pulses/s (at 25 °C [32]). *Neoconocephalus maxillosus* produces continuous calls with double-pulse rates of 130–145 pulses at 25 °C [32]. The carrier frequencies of both species overlap with those of *N. triops* that calls with ~11 kHz carrier frequency [47]. The pulse rates of the two sympatric species alone are highly attractive to *N. triops* females in Costa Rica at 25 °C. However, the weak attraction to continuous signals (Figure 1) should prevent or at least drastically reduce the chance of costly mismatings between species. In Puerto Rico, *N. maxillosus* occurs sympatrically with *N. triops* [32]. Here, the closed preference towards faster pulse rates should prevent *N. triops* females from approaching male *N. maxillosus*, despite the lack of a preference for versed calls (Figure 1B). Note that there are two potential scenarios to explain the difference in verse preference. Either a loss of selection in the Puerto Rican population led to the loss of the verse preference, or, selection by other species led to the gain of the verse preference in Costa Rica. Additional data would be necessary to distinguish between the two scenarios.

The differences reported in this study between populations in their preferences should allow for different evolutionary trajectories. For example, the Puerto Rican population could modify the verse structure (e.g., making longer verses) or lose verses altogether without losing attractiveness of their calls. Males of *N. triops*’ Florida population produce continuous calls through phenotypic plasticity when induced by adult diapause, while non-diapausing males produce versed calls [28,30]. Females of this population also lack the selectivity for verse structure and both call morphs are highly attractive to them [30]. Thus, the geographic variation in female preferences seemingly allowed *N. triops* with the Puerto Rican preferences to colonize temperate regions (i.e., Florida) by mitigating the consequences of the call structure plasticity.

Considering that *N. triops* is typically found in grasslands with similar habitat acoustics in Costa Rica and Puerto Rico and similar predators (i.e., bats, spiders, and parasitoid flies) are present throughout the tropical range (OMB personal observations), those factors have likely not played a role related to geographic differences in female preferences. Yet, whether the geographic differences in female preferences might be related to sexually preferred male traits remains to be tested. However, if sexual selection would explain the open preference in Costa Rican females at 25 °C, we would expect directional selection, i.e., an increase in phonotaxis scores with increasing trait values rather than the sigmoidal shape of the preference function.

## 5. Conclusions

Studying female mating preferences is essential for understanding the evolution of communication systems [48]. Our data provide a potential solution to the evolutionary paradox of how a communication system could allow for variation that does not disrupt the match between sender and receiver [4]. The lack of preferences (for call structure in Puerto Rican females) or wide preference functions (for pulse rate in Costa Rican females) on the receiver’s side allow for such variation in signals, and theoretically, subsequent changes in preferences through drift, selection, or mutations toward those new signal variants could lead to the diversification of the communication system [49]. We are not aware of comparable studies describing this kind of substantial geographic variation in female preferences in a reproductive communication system.

The range of *N. triops* covers most of the neotropics and extends north- and southward into neotropical habitats, reaching as far north as Ohio [28,32]. Geographic variation in signals and receiver preferences may have supported the spread into new habitats, as suggested above for expansion into temperate southeastern USA. Understanding how the communication system of *N. triops* has evolved towards the south may provide further insights into the role of geographic variation.

In the bigger picture, understanding how species, in general, and communication, in particular, evolves to new environmental conditions as the result of range expansion may provide comparable insights into how species respond to climate change. In this sense, studying geographic variation in various traits also informs about the underlying genetic and phenotypic potential of animals coping with novel environmental conditions [50,51] and more data of this kind are needed to estimate evolutionary ramifications of climate change.

## Figures and Tables

**Figure 1 biology-13-01026-f001:**
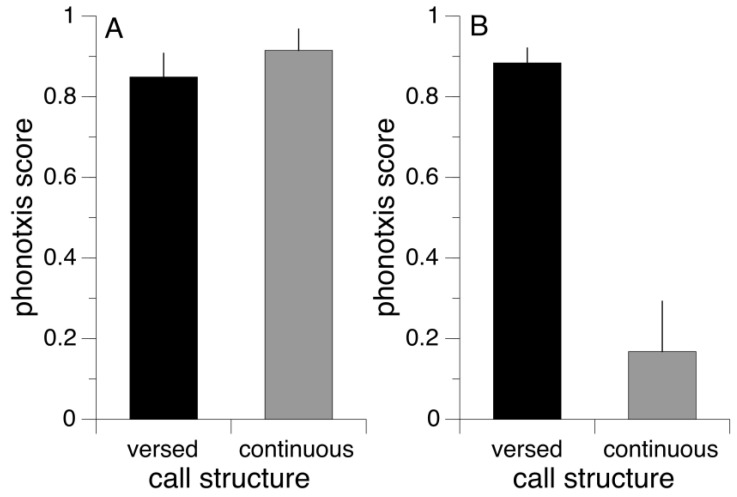
Average phonotaxis scores (± SEM) of female *N. triops* from (**A**) Puerto Rico (*N* = 9) and (**B**) Costa Rica (*N* = 10) to call stimuli structured in verses (black bars) or continuous (gray bars).

**Figure 2 biology-13-01026-f002:**
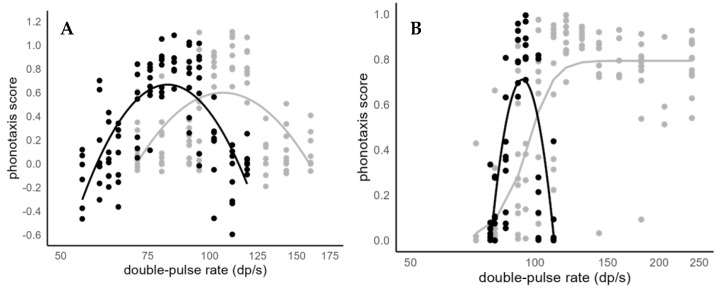
Phonotaxis scores of female *N. triops* from (**A**) Puerto Rico and (**B**) Costa Rica in response to varying double-pulse rate tested at 20 °C (black symbols and lines) and 25 °C (gray symbols and lines). The lines indicate the best-fit lines (see text) for the data distributions at each temperature.

**Table 1 biology-13-01026-t001:** Fit of phonotaxis scores of Puerto Rican females tested at 20 °C to linear, quadratic and sigmoidal models. ‘dAIC’ indicates the difference Akaike information criterion scores from the model with the lowest score, which is set to ‘0.00’ here, ‘DF’ for degrees of freedom, and ‘RSE’ for residual standard error. Asterisk indicates model with the best fit for each function.

Model (PR 20)	dAIC	DF	RSE
Linear	67.52	3	0.43
Quadratic *	0.00	4	0.31
Sigmoidal	69.50	4	0.43

**Table 2 biology-13-01026-t002:** Fit of phonotaxis scores of Puerto Rican females tested at 25 °C to linear, quadratic and sigmoidal models. ‘dAIC’ indicates the difference Akaike information criterion scores from the model with the lowest score, which is set to ‘0.00’ here, ‘DF’ for degrees of freedom, and ‘RSE’ for residual standard error. Asterisk indicates model with the best fit for each function.

Model (PR 25)	dAIC	DF	RSE
Linear	30.50	3	0.39
Quadratic *	0.00	4	0.33
Sigmoidal	11.88	4	0.35

**Table 3 biology-13-01026-t003:** Fit of phonotaxis scores of Costa Rican females tested at 20 °C to linear, quadratic and sigmoidal models. ‘dAIC’ indicates the difference Akaike information criterion scores from the model with the lowest score, which is set to ‘0.00’ here, ‘DF’ for degrees of freedom, and ‘RSE’ for residual standard error. Asterisk indicates model with the best fit for each function.

Model (CR 20)	dAIC	DF	RSE
Linear	60.85	3	0.39
Quadratic *	0.00	4	0.25
Sigmoidal	42.89	4	0.34

**Table 4 biology-13-01026-t004:** Fit of phonotaxis scores of Costa Rican females tested at 25 °C to linear, quadratic and sigmoidal models. ‘dAIC’ indicates the difference Akaike information criterion scores from the model with the lowest score, which is set to ‘0.00’ here, ‘DF’ for degrees of freedom, and ‘RSE’ for residual standard error. Asterisk indicates model with the best fit for each function.

Model (CR 25)	dAIC	DF	RSE
Linear	53.59	3	0.33
Quadratic	23.95	4	0.29
Sigmoidal *	0.00	4	0.26

## Data Availability

The raw data supporting the conclusions of this article will be made available by the authors on request.

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
