# Peer review of "Geographic Variation in Signal Preferences in the Tropical Katydid Neoconocephalus triops"

_biology, 2024, doi:10.3390/biology13121026_

Round 1
Reviewer 1 Report
Comments and Suggestions for Authors
In this manuscript, the authors showed female preferences for male calling signals in two geographically isolated populations and found these populations exhibited different selectivity for pulse rate and call structure. The methods and results were solid and interpretations were reasonable. I only have some minor comments, which, in my opinion, may help to improve the manuscript.
L115-116: Did the females used in this experiment experience the male calling signals before the test or were they prevented from any interspecific communication? I wonder if their phonotactic behavior might modified by their former experiences, which often happens in crickets.
L144-147: If I understood correctly, a long pulse was used for the stimulus instead of a double-pulse, right? How long was the duration of the long pulse?
L151: Probably the lower limit of the pulse rate was wrong. For the experiment for the PR population in 20 deg. I can see the data below 71.4 Hz.
L163-166: It is unclear how many different stimuli were presented to one female and how many individuals were tested in total. It is also not described whether the same individuals were used in both experiments in 20 and 25 deg. Were Individual differences taken into account for the data-fitting?
L191: What is this a, b, c mean? They are not defined before.
L221: and elsewhere: There is no A and B in Figure 2.
L309: 47] à [47]
Figure 2:
1) Please add A and B. (see the comment above)
2) “Double-pulse rate” should be “pulse rate”? (see L147)
3) This is not necessary but I think it is better to add some legends in the graphs. It must be helpful for readers.
Reviewer 2 Report
Comments and Suggestions for Authors
Beckers and Schul explored the geographic variation in female mating preferences of the katydid Neoconocephalus triops between populations from Puerto Rico and Costa Rica. Phonotaxis experiments with females tested their preference for pulse rate and call structure at different ambient temperatures. The study reports significant divergences between the two populations: in Puerto Rico, females expressed closed preferences for pulse rates, whereas in Costa Rica, females showed open preferences, particularly at higher temperatures. The authors attributed divergences to the existence of other sympatric katydid species that may exert some evolutionary pressures on communication systems. Methods were based on behavioral experiments and the analyses of signals that fit statistical models explaining phonotaxis scores. In that way, their results imply that reduced selectivity may allow variation in signals, which in turn may be one of the driving forces in the evolution of communication systems. These results add to the understanding of how geographic variation in mating preference can lead to diversification and speciation in communication systems.
The manuscript is based on a rather sound experimental design; however, there are concerns regarding the statistical analyses, signal processing, and interpretation of the results.
The statistics are too simplistic for using two-sample t-tests to compare phonotaxis scores between Puerto Rico and Costa Rica females. In fact, given that the data are multivariate (complex) on many variables—including temperature and pulse rates—the statistical analysis would be more rigorous if interactions were allowed, e.g., ANOVA or mixed-effects models. Linear, quadratic, and sigmoidal models will also be appropriate. However, basing the choice of model on simply the Akaike information criterion lacks depth. Support from residual analyses or validation with extra subsets will add weight to the strength of the model.
For signal analysis, the combination of double pulses into long pulses needs further validation. In their way, the data are easier to handle, but very small changes in pulse structure may be lost that could affect female preferences. While equalizing the amplitude of the stimuli, possible effects of a small degree of amplitude variation on phonotaxis scores are not considered, which may introduce bias, especially in species with very sensitive auditory systems.
I read with interest Triblehorn and Schul (2009); J Neurophysiol. 102(3):1348–1357. doi: 10.1152/jn.91276.2008. The authors found in preliminary tests that female N. triops were equally attracted to stimuli with merged long pulses compared to the double-pulse structure. This result indicates that the temporal repetition rate of the pulses (AM rate) is the relevant parameter rather than the exact pulse structure.
Results in Triblehorn and Schul (2009) thereby effectively address the first part of my comment about the merged pulses. The second might still be valid - about amplitude variation.
The results are well presented but represent an overinterpretation. For example, the significant variation at 25 ºC in response to pulse rates between populations is explained by evolutionary pressures. But with the total lack of physiological or neurological data, such a claim is weakened. Apart from evolutionary adaptation, other reasons may exist for the open preference at higher pulse rates in Costa Rican females, including plasticity in sensory processing under different environmental conditions.
The inferences about geographic variation are plausible but speculative in the absence of direct evidence about fitness consequences and/or mechanisms of reproductive isolation. This could be one intriguing idea that signal variation might facilitate speciation, but as it stands, this is a hypothesis. The implications are overstated, and this should be written in more conservative language.
Generally, the manuscript is well organized, but it is a methodologically conservative study. My main concerns, regarding both statistical interpretation and overconfidence in results, would be better dealt with by more nuanced approaches to modeling and hypothesis testing.
I would give the manuscript a score of 7.8 out of 10. It can be published with only moderate changes, but it needs to be more critical of the statistical rigor and data interpretation.
Line 73: "highly" instead of "strongly"
Line 163: "producing" instead of "eliciting"
Line 180: "phonotaxis" instead of "responses"
Line 222: "suggesting" instead of "indicating"
Line 240: "remained" with "persisted"
Line 292: "are" with "may be"
Line 351: "facilitated" with "supported"
